# KI04 an Aminoglycosides-Derived Molecule Acts as an Inhibitor of Human Connexin46 Hemichannels Expressed in HeLa Cells

**DOI:** 10.3390/biom13030411

**Published:** 2023-02-22

**Authors:** Cheng-Wei T. Chang, Naveena Poudyal, Daniel A. Verdugo, Francisca Peña, Jimmy Stehberg, Mauricio A. Retamal

**Affiliations:** 1Department of Chemistry and Biochemistry, Utah State University, Logan, UT 84322-0300, USA; 2Laboratorio de Neurobiología, Facultad de Medicina y Facultad de Ciencias de la Vida, Instituto de Ciencias Biomédicas, Universidad Andres Bello, Santiago 7780272, Chile; 3Center for Membrane Protein Research, Department of Cell Physiology and Molecular Biophysics, Texas Tech University Health Sciences Center, Lubbock, TX 79430-6551, USA; 4Programa de Comunicación Celular en Cáncer, Facultad de Medicina Clínica Alemana, Universidad del Desarrollo, Santiago 7610496, Chile

**Keywords:** connexins, hemichannels, inhibitors, aminoglycosides, lens, cancer

## Abstract

Background: Connexins (Cxs) are proteins that help cells to communicate with the extracellular media and with the cytoplasm of neighboring cells. Despite their importance in several human physiological and pathological conditions, their pharmacology is very poor. In the last decade, some molecules derived from aminoglycosides have been developed as inhibitors of Cxs hemichannels. However, these studies have been performed in *E. coli*, which is a very simple model. Therefore, our main goal is to test whether these molecules have similar effects in mammalian cells. Methods: We transfected HeLa cells with the human Cx46tGFP and characterized the effect of a kanamycin-derived molecule (KI04) on Cx46 hemichannel activity by time-lapse recordings, changes in phosphorylation by Western blot, localization by epifluorescence, and possible binding sites by molecular dynamics (MD). Results: We observed that kanamycin and KI04 were the most potent inhibitors of Cx46 hemichannels among several aminoglycosides, presenting an IC_50_ close to 10 μM. The inhibitory effect was not associated with changes in Cx46 electrophoretic mobility or its intracellular localization. Interestingly, 5 mM DTT did not reverse KI04 inhibition, but the KI04 effect completely disappeared after washing out KI04 from the recording media. MD analysis revealed two putative binding sites of KI04 in the Cx46 hemichannel. Results: These results demonstrate that KI04 could be used as a Cx46 inhibitor and could help to develop future selective Cx46 inhibitors.

## 1. Introduction

Connexins (Cxs) are a family of proteins located mostly at the plasma membrane, where they form two types of channels: hemichannels and gap junction channels (GJCs) [1]. While hemichannels are formed by the docking of six Cx-subunits and allow the flux of molecules and ions between the cytoplasm, the extracellular media, and vice versa [2], GJCs allow the exchange of molecules and ions between the cytoplasm of neighboring cells [1]. Both hemichannels and GJCs have been shown to play important physiological roles [3,4,5], and their dysfunction is associated with several pathologies [6,7]. There are 21 genes coding for different types of Cxs, but despite this heterogeneity, all Cxs share a common topology, including N- and C-termini located in the cytoplasmic side, four transmembrane domains (TM1-4), one intracellular loop (IL-1), and two extracellular loops (EL1-2) [8,9]. In all cases, the C-terminal is the most variable region and comprises several post-translational sites, which include sites for phosphorylation [10,11,12], redox- potential-mediated modifications [13,14,15], cleavage [16], and protein–protein interactions [17,18,19].

One of the main problems with the study of the roles of Cxs in both physiology and pathology is the lack of specific inhibitors/activators. Traditionally, carbenoxolone, heptanol, octanol, halothane, fatty acids, flufenamic acid, and molecules derived from glycyrrhetinic acid, to name a few, have been used to block both Cx hemichannels and GJCs [20]. Unfortunately, none of those molecules are selective for a specific type of Cx and/or usually affect other proteins, making their use in humans unsafe [20]. Thus, it is critical to develop new Cx modulators that can act as selective inhibitors with the potential for clinical use. In this context, a new group of hemichannel inhibitors derived from aminoglycosides has been recently developed [21]. Aminoglycosides are a family of small molecules with antibacterial activity, which include kanamycin, gentamicin, neomycin, and geneticin, among others. It was demonstrated in 2017 that a large group of aminoglycosides show inhibitory activity on Cx26, Cx43, and Cx46 hemichannels expressed in LB2003 *E. coli*, where the maximal potency was observed for geneticin on Cx26 hemichannels with an EC_50_ of 0.4 μM [22]. Although aminoglycosides have been used for several decades as antibiotics for the treatment of bacterial infections in humans, they can cause hearing loss and hepatotoxicity when used for prolonged periods of time or in high doses. For this reason, a set of aminoglycoside-derived molecules without antibiotic activity and reduced cytotoxicity [21] but retaining hemichannel blocking activity was developed [23]. Although promising, the effects of aminoglycoside-derived molecules on Cx hemichannel function were evaluated in transfected *E. coli*, which, compared to a mammalian cell, are much more simple in terms of size, type of metabolism, genome size, and therefore in their diversity of protein content, among many other characteristics [24]. Therefore, in order to obtain a specific aminoglycoside-derived inhibitor for a specific Cx type, it is necessary to study whether these molecules can act on hemichannels expressed in mammalian cells and then to determine which amino acids are crucial for the docking between these molecules and a given Cx.

Cx46 has been associated with cell survival in hypoxic conditions [25], and its dysfunction is associated with cataract formation [26]. Cx46 is also expressed in the cells of breast and brain cancer [27,28], where it seems to be associated with cancer aggressiveness [29]. As aminoglycosides have been shown to modulate Cx46 hemichannel activity when expressed in *E. coli* [22], they could potentially be used to treat Cx46-associated cataracts and cancer types. In this work, we studied whether KI04 [21], a kanamycin-derived molecule, could modulate Cx46 hemichannel activity when expressed in HeLa cells, to identify which region within the protein interacts with KI04.

We found that geneticin has no effects on Cx46 hemichannels, amikacin has a mild blocking effect, while both kanamycin and KI04 act as strong inhibitors of Cx46 hemichannels in a dose-dependent manner. Moreover, KI04 seems to bind to two putative segments of the Cx46 protein. Interestingly, one of them is the segment that links the TM1 with the EL1, which has been proposed to play an important role in hemichannel gating control.

## 2. Materials and Methods

### 2.1. Chemicals

Fluoromount-G was purchased from Electron Microscopy Science (Ft. Washington, PA, USA). Dithiothreitol (DTT) was obtained from Sigma-Aldrich (St. Louis, MO, USA). The rabbit monoclonal anti-Cx46 antibody was purchased from Santa Cruz Biotechnology (Santa Cruz, CA, USA).

### 2.2. Cell Culture and Transfections

HeLa cells (ATCC, Rockville, MD, USA) were cultured in DMEM with 10% fetal bovine serum (FBS), 100 U/mL penicillin, and 100 μg/mL streptomycin sulfate (Nunc, Roskilde, Denmark). Attached cells were dissociated for subculturing with 0.05% trypsin-EDTA (Thermo Fisher Scientific, Waltham, Massachusetts, USA). For transfection, HeLa cells were seeded in 6-well plates, and 24–48 h later (at ~60% confluence), they were transfected with 1 μg of pCMV6-AC-Cx46tGFP (turbo green fluorescent protein) vector (hCx46, Origene, Rockville, MD, USA), or pCMV6-AC-GFP for GFP expression only (Origene, Rockville, MD, USA). Briefly, Opti-MEM medium (Thermo Fisher Scientific, Waltham, MA, USA), plus Lipofectamine 2000 and 1 μg of the plasmids were mixed. Then, the cell culture medium was replaced with Opti-MEM, and the transfection mixture was added. After 4 h, cells were incubated in DMEM with 10% FBS at 37 °C, in 5% CO_2_, and 48 h later, cells were evaluated for Cx46 expression in an epifluorescence microscope. Selection started by adding the antibiotic G418 to the culture media 3 times per week for 2 weeks.

### 2.3. Dye Uptake

Hemichannel activity was evaluated through the uptake of DAPI at a final concentration of 10 μM. HeLa cells were grown in glass coverslips, and on the day of the experiments, a single coverslip was transferred to a 30 mm plastic dish and washed twice with control recording solution (in mM): 140 NaCl, 4 KCl, 2 CaCl_2_, 1 MgCl_2_, 5 glucose, and 10 HEPES, pH = 7.4 or a divalent cation-free solution (DCFS), which comprised (in mM): 140 NaCl, 4 KCl, 5 mM EGTA (an extracellular Ca^2+^ chelator), 5 glucose, and 10 HEPES, pH = 7.4. Changes in DAPI fluorescence intensity were evaluated in images taken every 20 s during a 20 min period at room temperature, using an inverted microscope (Eclipse Ti-U, Nikon). NIS Elements Advanced Research software (version 4.0, Nikon) was used for data acquisition and image analysis. The fluorescence intensity of 16 cells per experiment was averaged and plotted against time, and the slope of the linear rate of fluorescence increase, calculated with GraphPad Prism software version 9, was used as an indicator of the rate of dye uptake.

### 2.4. Microscopy Images

HeLa cells were grown on top of 12 mm glass coverslips until they reached 60% confluence. Then, they were exposed to KI04 for 20 min and washed with recording solution 3 times. Cells were fixed using 4% paraformaldehyde at room temperature for 10 min and were permeabilized using PBS with 0.1% Triton ×100 for 5 min at room temperature. Cells were finally washed with PBS and mounted in glass slides with Fluoromont-G containing DAPI. Mounted coverslips were placed in a dry and dark chamber for 24 h at 4 °C. Then, cells were observed under an inverted microscope (Eclipse Ti-U, Nikon) using adequate wavelengths to observe GFP and DAPI. Three pictures from three different fields for each cover were taken.

### 2.5. Western Blot

HeLa cells were lysed in PBS supplemented with protease inhibitors (Roche) and sonicated on ice. The protein concentration in the cell homogenate was determined using a Qubit protein assay kit (Thermo Fisher Scientific, Waltham, MA, USA) and read in a Qubit 3.0 fluorometer (Thermo Fisher Scientific, Waltham, MA, USA). Fifty micrograms of total protein was loaded in a 10% NuPAGE^TM^ gel (Thermo Fisher Scientific, Waltham, MA, USA) and electrotransferred to a PVDF membrane using an iBlot gel transference device (Thermo Fisher Scientific, Waltham, MA, USA). Membranes were incubated with an anti-Cx46 (Santa Cruz Biotechnology; 1:500) overnight at 4 °C. The next day, membranes were washed three times using TRIS buffer, pH 7.4, plus 0.05% Triton ×100. Finally, membranes were incubated with the secondary antibody conjugated with horseradish peroxidase (HRP) (Abcam) for 1 h at room temperature and were washed three times using TRIS buffer, pH 7.4, plus 0.05% Triton ×100. Proteins were detected using the Immobilon Forte Western HRP Substrate (Millipore, Burlington, MA, USA) and visualized with the LI-COR C-Digit Chemiluminescence Western Blot Scanner systems (LI-COR, Inc., Lincoln, Dearborn, MI, USA).

### 2.6. Whole-Cell Electrophysiology

HeLa cells were grown in a 60 mm plastic dish until 60–70% confluence. On the day of the experiment, cells were washed with recording solution twice, 1 mL of 0.05% trypsin-EDTA (Thermo Fisher Scientific, Waltham, MA, USA) was added, and cells were placed at 37 °C for approximately 1 min or until they appeared partially detached. At this point, cells were gently detached using a 1 mL micropipette. Cells were placed in a 15 plastic tube containing 5 mL DMEM with 10% FBS and centrifuged for 3 min at 1300 rpm. Then, the supernatant was discarded, and cells were resuspended in 6 mL of recording medium and centrifuged at 1300 rpm for 3 min twice. Finally, cells were resuspended in 200 μL of the recording solution and placed in a 1.5 mL Eppendorf tube. Following a recovery period (30 min at room temperature), cells were gently resuspended, and 8 μL of cell suspension was placed in Patchliner (NPC-16 Patchliner system, Nanion Technologies GmbH, Munich, Germany). Patch-control HT software (HEKA Elektronik, Lambrecht, Germany) was used to control the pressure necessary to establish the whole-cell patch clamp configuration. Hemichannel currents were recorded at room temperature (22–23 °C), using as internal solution (in mM): 10 NaF; 110 CsF; 20 CsCl; 2 EGTA; and 10 HEPES, pH 7.4 (adjusted with CsOH); and as external solution, the recording solution described above was used. Following cell contact with a 3–5 MΩ planar electrode, 30 μL of seal enhancer solution (in mM: 80 NaCl; 3 KCl; 35 CaCl_2_; 10 HEPES/NaOH, pH 7.4) was added to the external solution to promote giga-ohm seal formation. After establishing the whole-cell configuration, the seal enhancer solution was washed out with recording solution twice. Cells were allowed to stabilize for 2 min after starting whole-cell recordings. Patchmaster software (HEKA) was used to automatically compensate for whole-cell capacitance and series resistance and perform voltage-clamp protocols. Igor Pro 9 was used to analyze the data and create the figures.

### 2.7. Molecular Docking

A structural model of connexin46 (Cx46) was generated from the amino acid sequence available in the UniProt database (UniProt ID Q9Y6H8) using the SwissModel tool (https://swissmodel.expasy.org, accessed on 21 January 2023), taking the crystallized structure 7JKC as a template. The C-terminal domain was not included, as it has not been fully resolved in the crystal structure. The three-dimensional structures of the amikacin, geneticin, and kanamycin molecules were obtained from the PubChem database (https://pubchem.ncbi.nlm.nih.gov, accessed on 21 January 2023). Finally, the structure for the Ki04 molecule was constructed using Maestro 13.1 2021-1 software in its free academic version. The structures of the ligands, as well as those of Cx46 and the Cx46 hemichannel, were loaded into the PyRx software, and the protein structures were converted to PDBQT format for use in Autodock 4.2.6 software. A 200-step minimization was performed using a universal force field (UFF). A grid was configured so that the docking program utilized the entire surface area of both the Cx46 and the Cx46 hemichannel. Then, exploratory docking was performed where, in the first instance, Cx46 alone was used. Subsequently, a second exploratory docking was performed using the whole structure of the Cx46 hemichannel. In both procedures, a semi-flexible docking protocol was employed, keeping the protein domains rigid while the ligand remained flexible. Visualization of the data obtained was performed in Maestro 13.1 2021-1 software in its free academic version.

### 2.8. Statistical Analyses

Results are expressed as means ± SEM, and “n” refers to the number of independent experiments. For statistical analysis, each treatment was compared to its respective control, and significance was determined using a one-way ANOVA or paired Student’s t tests, as appropriate. Differences were considered significant when *p* < 0.05.

## 3. Results

### 3.1. HeLa Cells Only Showed Hemichannel Activity When Transfected with Cx46tGFP

HeLa cells are a cell line derived from a patient with cervical uterine cancer. This cell line has been used for many decades in Cx research because it lacks functional Cx-derived channels (both hemichannels and GJCs) [30,31,32]. In our previous work, we demonstrated that parental HeLa cells (non-transfected) do not form functional hemichannels, showing a very low basal rate of DAPI uptake, compared to Cx46-transfected cells [31]. In the present work, we transfected our parental HeLa cells with Cx46-GFP (Cx46tGFP) and with GFP alone as a transfection control. HeLa cells transfected only with GFP express the protein homogenously distributed in their cytoplasm and do not show DAPI uptake, either in control conditions or in DCFS (recording solution without divalent cations Ca^2+^ and Mg^2+^), which increases the Cx46 hemichannel opening probability [33] (Figure 1a,b). The Cx46tGFP-transfected cells under control conditions showed almost no DAPI uptake after 20 min of incubation (Figure 1a, DAPI, upper panels). However, when they were placed in DCFS, they showed a notorious increase in DAPI uptake (Figure 1a,b). These data show that transfection and selection procedures do not induce the appearance of functional hemichannels and demonstrate that Cx46 hemichannels are largely permeable to DAPI, making our cellular model appropriate for the study of the effect of putative Cx46 inhibitors.

### 3.2. KI04, a Kanamycin-Derived Aminoglycoside, Acts as Inhibitor of Cx46 Hemichannels

HeLa cells expressing Cx46tGFP were exposed to increasing concentrations of aminoglycosides in a range of 0–100 μM, and the rate of DAPI uptake was determined through time-lapse experiments. In this work, we studied the effect of geneticin, amikacin, kanamycin, and KI04. KI04 is a kanamycin-derived aminoglycoside in which 6′-NH_2_ was acylated with lauric acid (Figure 2a). Exposure to geneticin increased the rate of DAPI uptake (Figure 2b). However, this increase was not dose dependent, as the increase at any concentration tested was not statistically different from the rest, showing an average increase close to 60% (*n* = 4 independent concentration curves). Exposure to amikacin did not induce statistically significant changes in the rate of dye uptake (Figure 2c), although a small (~15%) increase in the rate of dye uptake was observed for low concentrations of 0.01 and 0.1 μM and a small (~22%) decrease at the higher concentrations of 10 and 100 μM. Neither the increase nor the decrease in DAPI uptake induced by amikacin was statistically different from the control (0 μM amikacin). In contrast, kanamycin showed a dose-dependent inhibition in the rate of DAPI uptake (Figure 2d), reaching ~73% inhibition at 10 μM and ~75% of inhibition at 100 μM, both of which were statistically different compared to the control (Figure 2d). The IC_50_ calculated for kanamycin was 1.7 μM. Similarly, KI04 also induced a dose-dependent decrease in the rate of DAPI uptake, with an IC_50_ for KI04 close to 7.7 μM.

### 3.3. The KI04 Inhibitory Effect Was Not Associated with Changes in Cx46 Molecular Weight or in Its Cellular Distribution

It is well accepted that changes in molecular weight (MW) in Cxs are associated with changes in phosphorylation [34]. In Western blots, the unphosphorylated form of Cx46 is associated with a 46 kDa band, while serine phosphorylation is associated with a 56–60 kDa band [35,36], and in general, hemichannel phosphorylation is associated with an inhibition in its function [37]. As aminoglycosides can induce protein kinase activation [38,39], we explored the possibility that KI04 induces its inhibitory effect via the phosphorylation of Cx46. Hence, we evaluated whether the effect of KI04 could be mediated by changes in the MW of Cx46. HeLa cells transfected with Cx46tGFP were exposed for 20 min to KI04 in different concentrations ranging between 0 and 100 μM, and changes in MW were analyzed by Western blot (WB). No obvious changes in band migration were observed (Figure 3a), suggesting that Cx46 does not present changes in phosphorylation after KI04 exposure. In addition, KI04 did not induce significant changes in total Cx46 levels compared to controls (Figure 3a). Then, we studied possible KI04-induced changes in Cx46 distribution by analyzing tGFP localization. After 20 min of exposure to 10 μM KI04, no obvious changes in Cx46tGFP distribution were observed. Under control conditions, Cx46tGFP cells showed most of their GFP fluorescence located near or surrounding the nucleus, although small dots could be observed in the cytoplasm and a faint signal at the edge of the plasma membrane (Figure 3b, control). A similar distribution was observed after KI04 (Figure 3b, KI04). Aminoglycosides can induce oxidative stress [40], and our group has demonstrated that Cx46 hemichannels are sensitive to changes in redox potential [14], where oxidations in the extracellular cysteines seem to induce Cx46 hemichannel closure [13]. Therefore, we then explored the possibility that the inhibitory effect of KI04 could be reversed by a cysteine-reducing agent. Unexpectedly, the addition of 5 mM DTT to the bath solution increased even further the inhibition induced by 10 μM KI04. Thus, in this set of experiments, 10 μM KI04 induced ~80% inhibition of the rate of DAPI uptake, and after DTT was added, the inhibition reached ~97% (Figure 3c). However, we observed that 5 mM DTT under control conditions induced a nonstatistical ~17% reduction in the rate of DAPI uptake. These results suggest that the inhibitory effect of KI04 does not depend on the oxidation of cysteines. Finally, we studied the effect of washing out KI04 from the extracellular media. In order to do that, we exposed Cx46tGFP cells to 10 μM KI04 for 10 min and then washed it out. We observed that after washing out the KI04 from the recording media, the rate of DAPI uptake was not only recovered but also increased by ~53% Figure 3d) (*n* = 3).

### 3.4. KI04 Decreases Whole-Cell Cx46 Hemichannel-Mediated Currents

We studied whether KI04 can act as an inhibitor when Cx46 hemichannels are opened by changes in the plasma membrane voltage. To attempt to answer this question, we performed whole-cell patch clamp experiments. Under control conditions, a ramp that moved the voltage from −80 to +80 mV in a 20 s period induced an almost-linear current between −80 and +20 mV approximately (Figure 4A, black line). From +20 mV and above, an exponential relationship between voltage and current was observed. This exponential increase was decreased by ~65% after exposure to 10 μM KI04 (Figure 4A, red line), as the current was activated above +20 mV and was decreased by KI04 (from 1132 ± 97 to 740 ± 14 pA at +80 mV) (Figure 4B). We suggest that KI04 is able to not only inhibit Cx46 hemichannels when opened by the remotion of Ca^2+^ from the extracellular media but also when opened by increased voltage.

### 3.5. KI04 Shows Two Putative Binding Sites with Cx46 Hemichannels

To study a plausible mechanism by which KI04 inhibits Cx46 hemichannels, we performed a molecular docking simulation between Cx46 and KI04 to identify the domains in which KI04 may have a greater probability of interaction (Figure 5a). One of these sites was found in a segment that connects the TM1 and the EL1 (Figure 5b). In this putative binding site, KI04 may form hydrogen bonds with Asp43, Trp45, Gly46, and Asp47 of Cx46. Interestingly, in a previous publication, we demonstrated that Cys mutation to Ala interfered with the same segment that was correlated with the closing of Cx46 hemichannels [41]. The other binding site found was located inside of the Cx46 hemichannel pore (Figure 5c), where Ki04 could interact with Tyr168, Leu170, and Glu174.

## 4. Discussion

In this work, we demonstrate that kanamycin and the kanamycin-derived molecule (KI04) act as inhibitors of Cx46 hemichannels expressed in HeLa cells. Additionally, we found that kanamycin and KI04 inhibited Cx46 hemichannels in a dose-dependent manner (IC_50_ 1.7 and 7.7 μM, respectively), and this inhibition is reversible when the aminoglycoside is washed out. We also found that the inhibition of Cx46 hemichannels induced by KI04 does not depend on extracellular Cys or changes in phosphorylation. Thus, our results revealed that KI04 is a new Cx46 hemichannel inhibitor that could be used to block hemichannels in Cx46-expressing mammalian cells with an IC_50_ in the low μM range.

We used HeLa cells because they lack functional Cx-derived channels [30,31,32]. HeLa cells expressing only GFP showed the same low rate of DAPI uptake when placed in media with or without extracellular Ca^2+^ and Mg^2+^, suggesting that our HeLa cells do not have functional hemichannels formed by any other type of Cx. Only HeLa cells expressing Cx46tGFP showed an evident increase in DAPI uptake under DCFS, suggesting that our HeLa cells are a good model to study in detail the changes in DAPI uptake when Cx46 is expressed.

Previous studies have demonstrated that native aminoglycosides and their derivatives act as inhibitors of Cx26, Cx43, and Cx46 hemichannels when expressed in *E. coli* [22]. In a previous work, our group demonstrated that gentamicin can inhibit Cx26 hemichannels when expressed in HeLa cells [42]. Thus, it is likely that kanamycin and KI04 also affect Cx26 and Cx43 hemichannels expressed in mammalian cells. Whether KI04 shows any selectivity or greater affinity for a given Cx or to what extent it may affect Cx gap junction channels are issues that must be clarified in future studies. Here, we demonstrate that kanamycin and KI04 act as inhibitors on Cx46 hemichannels when expressed in HeLa cells. The other aminoglycosides evaluated did not show significant effects. KI04 has no antibiotic activity, and hence, it could be a good starting point for drug discovery to attain nonselective small molecules with the potential to be used to block Cx hemichannels in mammalian cells, either for basic and/or translational research.

HeLa cells transfected with Cx46 showed very weak gap-junction-mediated dye transfer using the scrape-loading technique (data not shown), so it was not possible to measure the potential effects of kanamycin and KI04 on Cx46 gap junction channel activity. This is an issue that will require the use of a different cell model that can express measurable gap junctional activity.

To investigate the difference in the Cx46 inhibition by traditional aminoglycosides and KI04, molecular docking was employed. We identified two possible sites of interaction between the KI04 and Cx46 hemichannels. Interestingly, one of these sites involved amino acids located in the interaction domain between TM1 and EL1, which has been postulated as a segment that controls permeability, voltage, and Ca^2+^ dependency [43,44]. Accordingly, the Gly45 in Cx30 and Cx43 [45] is directly involved in hemichannel Ca^2+^ sensing and, therefore, in hemichannel gating. Previously, we suggested that the chemical modifications of Cx46 extracellular Cys induce changes in this segment (between amino acids 47 and 61), which could be responsible for the closing of Cx46 hemichannels [41]. Moreover, mutations in this particular segment (also called the parahelix), especially in Gly45 in Cx31, are associated with *Erythrokeratoderma variabilis* [46] and in Cx26 is associated with keratitis–ichthyosis–deafness syndrome [47]. Similarly, mutations in Gly46 in Cx50 have been associated with cataract development [48], and mutations in positions 45 and 47 of Cx46 also have been associated with cataract formation [49]. In summary, the parahelix segment could be the main binding site for KI04 in Cx46 hemichannels. This can be attributed to the presence of the linear lipid chain on KI04, which manifests the differences in the mechanisms of inhibition as compared to traditional aminoglycosides. However, future experiments focused on verifying this hypothesis will be necessary.

## 5. Conclusions

KI04 represents a new class of Cx46 hemichannel inhibitors. KI04 inhibits Cx46 hemichannels expressed in mammalian cells in low μM concentrations through unique modes of inhibition. Therefore, KI04 opens an opportunity to develop specific Cx46 hemichannel inhibitors based on putative sites of interaction.

## Figures and Tables

**Figure 1 biomolecules-13-00411-f001:**
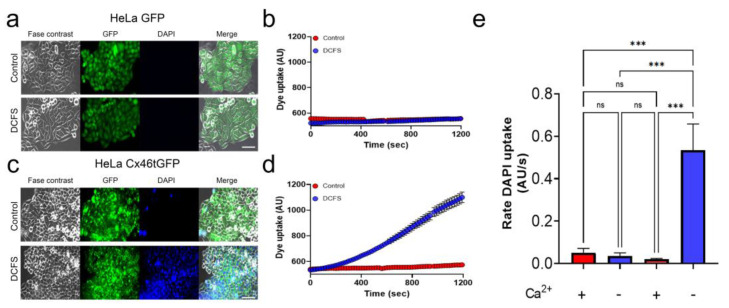
Only HeLa cells transfected with Cx46tGFP show DAPI uptake in DCFS. HeLa cells were transfected with GFP or Cx46tGFP and DAPI uptake time-lapse recordings were performed. (**a**) Pictures of a culture of HeLa cells expressing GFP (HeLa-GFP) revealed that almost all HeLa cells express this protein (merge panels). After 20 min of exposure to 10 μM DAPI, HeLa-GFP cells do not show DAPI uptake, either under control conditions or in DCFS (DAPI panel). (**b**) Representative quantification of DAPI uptake in time-lapse experiments showing that the rate of DAPI uptake in HeLa-GFP cells under control (red dots) and DCFS (blue dots) conditions is very similar. (**c**) HeLa cells transfected with Cx46tGFP (HeLa Cx46tGFP) do not show significant DAPI uptake in control conditions (DAPI panels). However, DAPI uptake was greatly increased when cells were placed in DCFS. (**d**) Representative quantification of DAPI uptake in time-lapse experiments showing a significant increase in DAPI uptake in HeLa Cx46tGFP in DCFS (blue dots) in comparison to the control condition (red dots). Each dot corresponds to the average intensity of DAPI fluorescence in the nucleus of 16 different cells. (**e**) Graph showing the average of *n* = 3 independent experiments for each condition (control and DCFS). *p* > 0.05 = n.s, and *p* < 0.001= ***.

**Figure 2 biomolecules-13-00411-f002:**
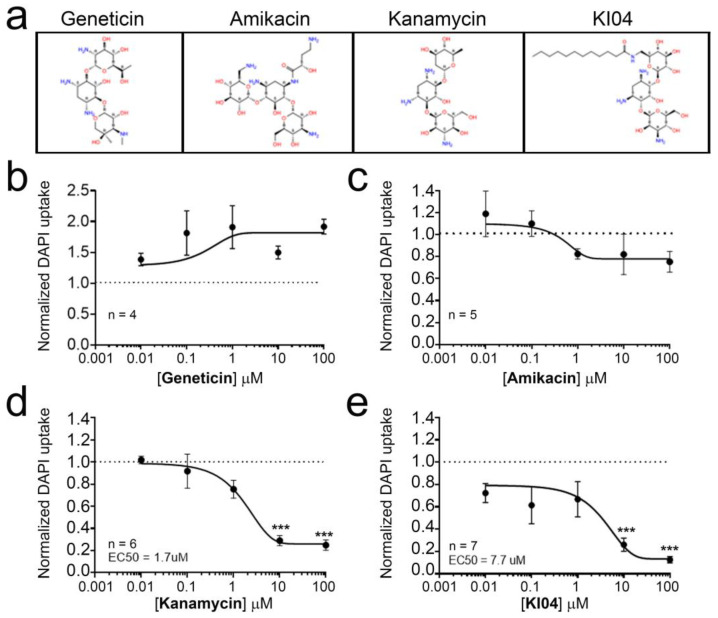
KI04 acts as an inhibitor of Cx46 hemichannels. HeLa cells transfected with Cx46tGFP were placed in DCFS complemented with 10 μM DAPI, and time-lapse experiments were performed. (**a**) Chemical structure of aminoglycosides and KI04 that were used in this work. Dose response of (**b**) geneticin (*n* = 4), (**c**) amikacin (*n* = 5), (**d**) kanamycin (*n* = 6), and (**e**) KI04 (*n* = 7) on DAPI uptake. In each plot, the rate of DAPI uptake was normalized against the rate of DAPI uptake measured in the absence of KI04 (dot line). For each experiment, DAPI fluorescence intensity was measured in 16 cells, showing GFP or Cx46tGFP fluorescence. *p* < 0.01 = ***.

**Figure 3 biomolecules-13-00411-f003:**
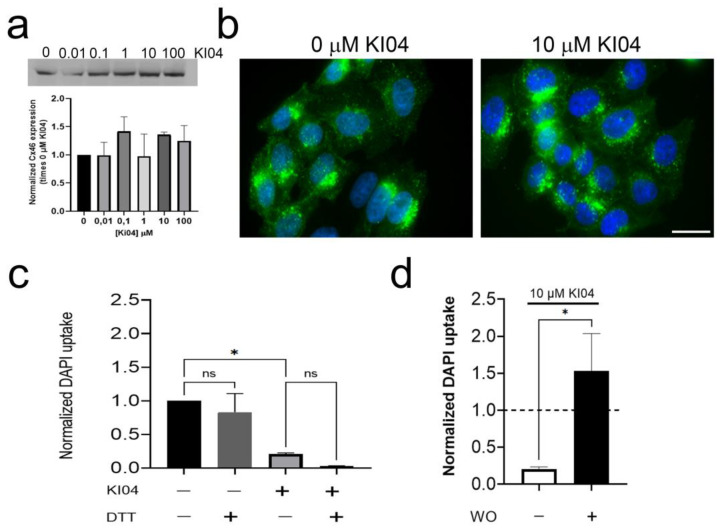
KI04 does not induce changes in Cx46 molecular weight and cellular distribution. (**a**) Western blot analyses show that none of the KI04 concentrations used in this study (from 0.01 to 100 μM) induced changes in Cx46 electrophoretic mobility. Similarly, (**b**) 10 μM KI04 did not induce evident changes in Cx46 intracellular localization. (**c**). Addition of 5 mM DTT shows a tendency to reduce the rate of DAPI uptake, and moreover, it did not reverse the inhibition mediated by KI04. (**d**) On the contrary, for Cx46tGFP cells that were transiently exposed to 10 μM KI04, the inhibitory effect was completely reversed, and moreover, there was an increase in DAPI uptake (*n* = 4 for each experiment, 16 cells analyzed in each case). For each experiment, DAPI fluorescence intensity was measured in 16 cells. *p* < 0.05 = * and no statistical differences (*p* > 0.05), was denoted as ns.

**Figure 4 biomolecules-13-00411-f004:**
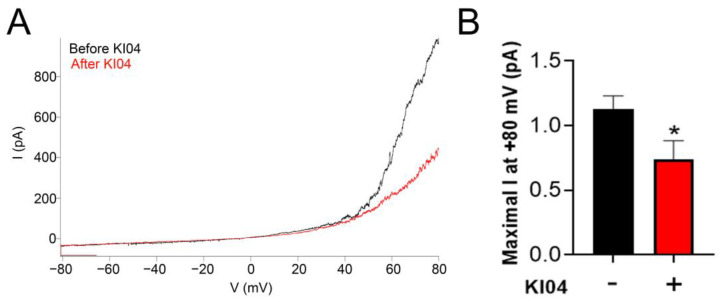
KI04 inhibits Cx46-mediated hemichannel currents. Hemichannel currents were studied in HeLa cells expressing Cx46tGFP. (**A**) Voltage ramps from −80 to +80 mV were applied in whole-cell patch clamp configuration. Large currents were recorded in HeLa Cx46tGFP cells that were activated above +20 mV. These currents were decreased when cells were exposed to 10 μM KI04. (**B**) Quantification of the maximal current at +80 mV in *n* = 3 independent experiments, *p* < 0.05 = *.

**Figure 5 biomolecules-13-00411-f005:**
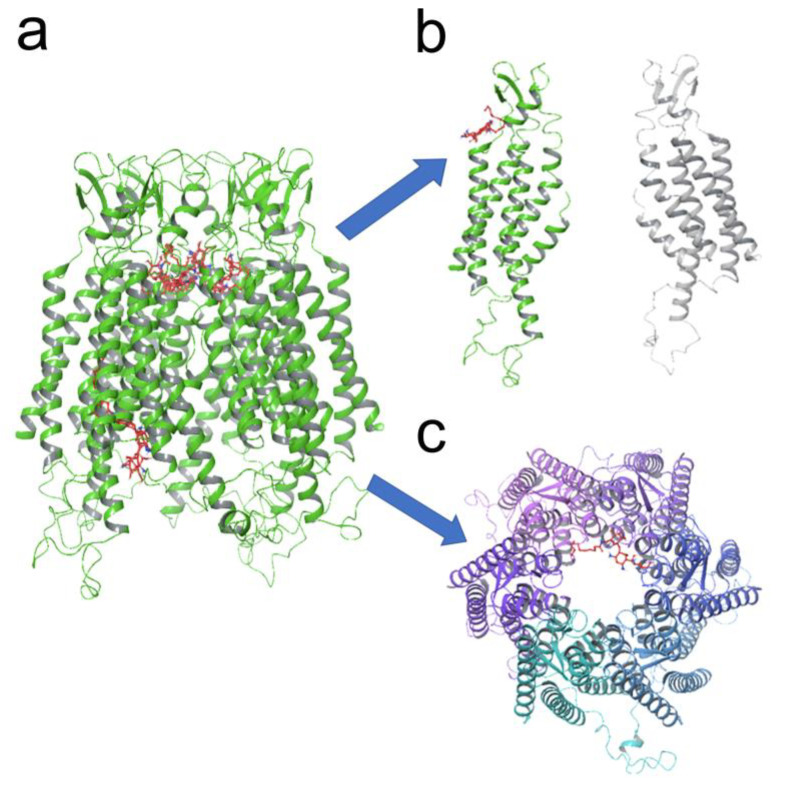
KI04 shows two putative binding sites with Cx46 hemichannels. Human Cx46 hemichannels were modeled in silico, and a molecular docking analysis was performed in the presence of KI04. (**a**) In our model, KI04 showed two putative binding sites, (**b**) one close to the parahelix segment and (**c**) another inside of the pore, close to the cytosolic entrance.

## Data Availability

All data are available as requested by the corresponding authors.

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
