# Peer review of "KI04 an Aminoglycosides-Derived Molecule Acts as an Inhibitor of Human Connexin46 Hemichannels Expressed in HeLa Cells"

_biomolecules, 2023, doi:10.3390/biom13030411_

Round 1

Reviewer 1 Report

In this manuscript, Chang et al. report that KI04, a kanamycin-derived molecule, potently inhibited Cx46 hemichannels activity in cultured HeLa cells transfected with the human Cx46tGFP. In general, the authors addressed an interesting topic. The manuscript is well-written and well-organized. Their observations support the conclusion. As for the weaknesses, the working mechanism of KI04 has yet to be explored. In addition, the potential use of KI04 in Cx46-related pathophysiological situations has not been tested.  

Major comments

1) It needs to be clarified whether the effect of KI04 is Cx46-specific. It could be interesting to test whether KI04 also affected hemichannel activity formed by other isoforms of connexins or pannexins. What about the putative binding sites of KI04 in different isoforms of Cxs? These points need to be discussed. 

2) The inhibitory effect of KI04 on Cx46 hemichannels disappeared after washing. Is KI04 a membrane-permeable chemical? What about the effect of KI04 on Cx46-formed intercellular gap junctional communication? 

Minor comments

1) The sentences in lines 76~77, "Because ..."; line 150, "the day of the ..." is incomplete. A minor grammatical check is necessary. 

Author Response

We wish to thank you all for your time and comments, which have contributed to improving our Manuscript. We have added a response to each comment. We have addressed all the issues raised by the Reviewers and hope the Manuscript is now deemed suitable for publication. We show at the end of each response, an excerpt of the respective text that was added to the Manuscript to ease your revision and have highlighted in yellow those changes within the Manuscript.

Reviewer Comments:

Reviewer 1: In this manuscript, Chang et al. report that KI04, a kanamycin-derived molecule, potently inhibited Cx46 hemichannels activity in cultured HeLa cells transfected with the human Cx46tGFP. In general, the authors addressed an interesting topic. The manuscript is well-written and well-organized. Their observations support the conclusion. As for the weaknesses, the working mechanism of KI04 has yet to be explored. In addition, the potential use of KI04 in Cx46-related pathophysiological situations has not been tested.  

Response: We thank the Reviewer for the positive opinions. In the future we plan to explore the mechanisms of action, by performing new molecular dynamics and dockings to identify the critical amino acids involved in the Cx46-KI04 interaction and generate Cx46 single-residue mutations to test them in vitro. Regarding, KI04´s potential use, we are currently evaluating KI04 in a cancer model that expresses Cx46. We expect to be publishing those results in a follow up publication. We also plan to identify a potential KI04-related “pharmacophore” and perform de novo small molecule design to hopefully develop Cx46 hemichannel inhibitor small molecules, with focus on their potential selectivity for Cx46.

Major comments

Reviewer 1: It needs to be clarified whether the effect of KI04 is Cx46-specific. It could be interesting to test whether KI04 also affected hemichannel activity formed by other isoforms of connexins or pannexins. What about the putative binding sites of KI04 in different isoforms of Cxs? These points need to be discussed. 

Response: We thank the Reviewer for this comment. As shown in a series of papers by Chang and Altenberg, kanamycine-derived molecules are not specific for (at least) Cx26, Cx43 and Cx46 hemichannels when expressed in E. coli. So, KI04 is likely to have effects on Cx26 and Cx43 hemichannels when expressed in Hela cells. This is supported by a previous publication of ours showing that gentamycin inhibits Cx26 hemichannels in Hela cells (Figueroa et al., 2014).

We focused on the effects of KI04 on Cx46 hemichannels, because we have a lot of experience working on this model and we have Cx46-related models of several types of cancers in which KI04 can be evaluated.  Throughout the Manuscript we tried to be careful not to suggest that KI04 is a specific Cx46 inhibitor, but that it may be a first step to find a specific Cx46 inhibitor. As mentioned in the answer above, we plan to follow this study and try to design a selective Cx46 hemichannel inhibitor that could potentially be used for the treatment of Cx46-hemichannel-associated pathologies such as cataracts and some types of cancers. Given the relevance of the point raised by the Reviewer, to further clarify this issue, we have added the following paragraphs:

In the abstract:

 “These results demonstrate that KI04 could be used as a Cx46 inhibitor and could help to develop future selective Cx46 inhibitors”.

In the discussion section:

“Previous studies have demonstrated that native aminoglycosides and their derivatives act as inhibitors of Cx26, Cx43 and Cx46 hemichannels when expressed in E. coli [22]. In a previous work our group demonstrated that gentamycin can inhibit Cx26 hemichannels when expressed in HeLa cells [42]. Thus, it is likely that Kanamycin and KI04 also affect Cx26 and Cx43 hemichannels expressed in mammalian cells. Whether KI04 shows any selectivity or greater affinity for a given Cx or to which extent if may affect Cx gap junction channels are issues that must be clarified in future studies. Here we demonstrate that kanamycin and KI04 act as inhibitors on Cx46 hemichannels when expressed in HeLa cells. The other aminoglycosides evaluated did not show significant effects. KI04 has no antibiotic activity and hence, it could be a good start for drug-discovery to attain non-selective small molecules with the potential to be used to block Cx hemichannels in mammalian cells, either for basic and/or translational research.”

2) The inhibitory effect of KI04 on Cx46 hemichannels disappeared after washing. Is KI04 a membrane-permeable chemical? What about the effect of KI04 on Cx46-formed intercellular gap junctional communication? 

Response: This a very interesting issue. When Cx46 is expressed in Hela cells, they form fully functional Cx46 hemichannels, but their gap junction channel-dependent intercellular coupling is very weak. Hence, it is very difficult to measure gap junctional-mediated coupling and it is even harder to evaluate an inhibitor like KI04. In consequence, HeLa cells are not the proper model to study the potential effects of KI04 on Cx46 gap junctional communication. Experiments to answer this very interesting question are required, using a different cell model that forms functional Cx46-gap junction channels with a measurable activity.

We have added the following sentence to the discussion sections dealing with this issue.

“HeLa cells transfected with Cx46 showed very weak gap junction-mediated dye transfer using the scrape-loading technique (data not shown), so it was not possible to measure the potential effects of Kanamycin and KI04 on Cx46 gap junction channel activity. This is an issue that will require que use of a different cell model that can express measurable gap junctional activity”.     

Minor comments

Reviewer: The sentences in lines 76~77, "Because ..."; line 150, "the day of the ..." is incomplete. A minor grammatical check is necessary. 

Response: We apologize for this mistake and thank the Reviewer for noticing these mistakes. The sentences in question were corrected and now read as follows:

“As aminoglycosides have been shown to modulate Cx46 hemichannel activity when expressed in E. coli [22], they could potentially be used to treat Cx46-associated cataracts and cancer types”.

“The day of the experiment, cells were washed with recording solution twice, 1 ml of 0.05% trypsin-EDTA (ThermoFisher Scientific, Massachusetts, USA) was added, and the cells were placed at 37°C for approximately 1 min or until cells appeared partially detached”.

Reviewer 2 Report

The author presents a study investigating the inhibitory effects of aminoglycosides on connexin 46 hemichannels expressed in HeLa cells and reports that KI04 exhibits significant dose-dependent activities and reduces hemichannel-mediated currents in whole cells. The manuscript is clear and the technical approach used is appropriate. Except section 3.3, where the author measures molecular weight through Western blot analysis, but fails to mention what could cause a change in molecular weight and its relationship to phosphorylation. If phosphorylation affects molecular weight, is it significant enough to be measured by Western blot?

Author Response

We wish to thank you all for your time and comments, which have contributed to improving our Manuscript. We have added a response to each comment. We have addressed all the issues raised by the Reviewers and hope the Manuscript is now deemed suitable for publication. We show at the end of each response, an excerpt of the respective text that was added to the Manuscript to ease your revision and have highlighted in yellow those changes within the Manuscript.

Reviewer 2: The author presents a study investigating the inhibitory effects of aminoglycosides on connexin 46 hemichannels expressed in HeLa cells and reports that KI04 exhibits significant dose-dependent activities and reduces hemichannel-mediated currents in whole cells. The manuscript is clear and the technical approach used is appropriate. Except section 3.3, where the author measures molecular weight through Western blot analysis, but fails to mention what could cause a change in molecular weight and its relationship to phosphorylation. If phosphorylation affects molecular weight, is it significant enough to be measured by Western blot?

 Response: We wish to thank the reviewer for this comment. It is well accepted that changes in molecular weight (MW) in Cxs are associated to changes in phosphorylation, and in general, hemichannel phosphorylation is associated to an inhibition of its function. As aminoglycosides can induce protein kinase activation, we explored the possibility that KI04 induces its inhibitory effect via phosphorylation of Cx46. We have recently analyzed the different Cx46 posttranslational modifications, among them phosphorylation, and found that Cx46 migrates in two main forms in a Western blot gel, one with 46 kDa, corresponding to its unphosphorylated form, and a 56-60 kDa band, which corresponds well to Cx46 with serine-phosphorylated residues. In the present study, we only observed a single band at 46 KDa, suggesting that KI04 did not induce its effects by inducing the phosphorylation of Cx46.

To clarify this issue, we have added the following paragraph at the results section:

 “3.3 The KI04 inhibitory effect was not associated to changes in Cx46 molecular weight or in its cellular distribution. It is well accepted that changes in molecular weight (MW) in Cxs are associated to changes in phosphorylation [34]. In Western blots, the unphosphorylated form of Cx46 is associated to a 46 kDa band, while serine phosphorylation is associated to a 56-60 kDa band [35,36], and in general, hemichannel phosphorylation is associated to an inhibition in its function [37]. As aminoglycosides can induce protein kinase activation [38,39], we explored the possibility that KI04 in-duces its inhibitory effect via phosphorylation of Cx46”.

Round 2

Reviewer 1 Report

The authors have adequately addressed my concerns. I have no additional comments. 

Reviewer 2 Report

Thanks for the clarification.